# Further Validation Study of the Gender-Specific Binary Depression Screening Version (GIDS-15) and Investigation of Intervention Effects

**DOI:** 10.3390/bs15091253

**Published:** 2025-09-14

**Authors:** Jan S. Pellowski, Christian Wiessner, Claudia Buntrock, Hanna Christiansen

**Affiliations:** 1Philipps University Marburg, Department of Psychology, 35032 Marburg, Germany; 2Institute for Sex Research, Sexual Medicine and Forensic Psychiatry, University Medical Center Hamburg-Eppendorf, 20251 Hamburg, Germany; 3Institute of Medical Biometry and Epidemiology, University Medical Center Hamburg-Eppendorf, 20246 Hamburg, Germany; 4Institute of Social Medicine and Health Systems Research, Medical Faculty, Otto-von-Guericke-University Magdeburg, 39120 Magdeburg, Germany; 5Philipps University Marburg, Department of Clinical Child and Adolescent Psychology, 35032 Marburg, Germany; 6Child and Adolescent Outpatient Clinic Marburg, 35032 Marburg, Germany; 7German Center for Mental Health, 35032 Marburg, Germany

**Keywords:** depression, male depression, gender-specific binary depression screening version, sensitivity to change, subclinical depression

## Abstract

Men and women differ in the manifestation of depression. At the same time, there is a lack of gender-sensitive depression questionnaires in Germany. This study investigated the Gender-specific binary depression screening version (GIDS-15) in a further validation step. In a two-armed, pragmatic single-blind randomised controlled clinical trial, we first investigated the psychometric properties and the sensitivity to change in the GIDS-15 in a sample with subclinical depression (*N* = 203). In addition, we then analysed sex differences between the intervention and waiting control group over time. We were able to demonstrate adequate to acceptable internal consistency as well as convergent construct validity of the GIDS-15. Additionally, we were able to demonstrate the sensitivity to change in the GIDS-15. Using a linear mixed model, we calculated a three-way interaction between intervention group, sex, and time (*p* = 0.017). We found an increase in the intervention effect for men over time. Conclusions: The GIDS-15 proves to be a solid and practical screening tool for the gender-sensitive assessment of depression in Germany. It can be used for progression and intervention diagnostics, although the intervention effect that was found can only be interpreted to a limited extent due to significant sample size differences between men and women. Limitations of our study and practical implications are discussed.

## 1. Introduction

The scientific evidence for a gender-specific expression of depressive symptoms has increased in recent years. While women with depression are more likely to report internalising symptoms consistent with conventional depression criteria compared to men, men with depression are more likely to report externalising symptoms than women ([12], [13]; [31]; [42]; [58]; [38]). At the same time, studies show that externalising symptoms are also reported by women with depression ([31]; [35]) and that men with depression also report internalising symptoms ([31]). Both internalising and externalising symptoms are therefore to be expected in both sexes. As an explanation, reference is made to the individual orientation towards social roles, expectations, identities, and forms of expression associated with social gender—in short: gender—as opposed to biological sex ([1]; [14]; [16]; [36]; [56]).

These research findings have since then led to the inclusion of an important addition on sex and gender differences in the phenomenology and course of depression in the text revision of the fifth edition of the Diagnostic and Statistical Manual of Mental Disorders (DSM-5-TR; [3]) ([43]). According to this text revision, women with depression tend to report somatic complaints, such as impaired appetite and sleep, or interpersonal sensitivity ([3]). In comparison, men with depression tend to have maladaptive self-coping and problem-solving strategies, such as substance use, risk-taking, or reduced impulse control ([3]). At the same time, this text revision cannot yet be equated with a change in diagnostic criteria ([43]), especially as only selected findings appear to be referenced, while others are not mentioned (e.g., [31]). In the recently introduced eleventh version of the International Statistical Classification of Diseases and Related Health Problems (ICD-11; [60]), irritability is also included as a possible affective component of depression in adults in the disorder description, which could facilitate the identification of depressive disorders in men by clinicians or in primary care ([37]). It should be noted that irritability was already listed in the DSM-IV as a major criterion for depression in children and adolescents, while it was not considered as such in adults ([2]).

Internationally, various measurement instruments have been developed and validated on the basis of research findings ([62]), such as the Gender Inclusive Depression Scale (GIDS; [31]), the Gotland Male Depression Scale (GMDS; [49]; [48]), the Masculine Depression Scale (MDS; [30]), or the Male Depression Risk Scale (MDRS-22; [42]). However, the measurement instruments differ in their design and the composition of the symptom areas recorded ([39]; [62]). For Germany, the Gender-specific binary depression screening version (GIDS-15; [39]) based on the GIDS ([31]) was recently analysed in two large mixed-sex samples in an initial validation study. After factor analysis, the screening procedure has a 5-factor solution (depressive symptoms, stress perception, anxiety, aggressiveness, and substance use) and consists of 15 items. The psychometric properties are generally satisfactory. Expected sex effects are largely demonstrated ([39]). In addition, the Gender-Sensitive Depression Screening (GSDS; [34]; [33]) is another measurement instrument from Germany, which in turn measures individual, different symptom areas compared to the GIDS-15. For example, while both methods measure aggression and thus correspond to the new disorder descriptions of DSM-5-TR ([3]) and ICD-11 (WHO), the GIDS-15 also measures anxiety ([39]). According to recent findings, anxiety plays a particular role in suicidality in men, which is highly associated with depression, and is, *inter alia*, linked to feelings of losing control ([23]; [57]; [43]). In addition, the GIDS-15 has fewer items than the GSDS and is therefore more economical to use.

There are considerations and demands to use screening instruments to record male depression symptoms, preferably alongside standardised procedures, in primary care and in longitudinal studies ([43]). In the long term, this is associated with the expectation of refining additional criteria for depression in men ([43]). Well-validated measurement instruments are essential for this refinement. As described above, the GIDS-15 has demonstrated to be promising with its initial findings ([39]). However, previous validation studies were often not validated on relevant samples and only on self-assessment instruments due to the lack of external clinical judgements ([39]). Furthermore, to the best of our knowledge, none of the above-mentioned measurement instruments have yet been analysed for their sensitivity to change, which is, however, an important precondition for their use in the context of follow-up diagnostics. To the best of our knowledge, there are also no findings to date on the use of the measurement instruments in controlled intervention study designs. In addition, the investigation of intervention effects between the sexes over time is important in order to derive important treatment aspects for clinicians. These could, for example, relate to the content of treatment, but also to the form of treatment. For example, it would be worth considering whether the sexes differ in their intervention outcomes if, for example, the aspect of maladaptive problem-solving strategies ([3]) is adequately addressed therapeutically. With regard to the form of intervention, adherence-focused guidance concepts in online formats could be helpful ([32]).

### Aims of the Current Study

This study addresses the aforementioned considerations and, in a further validation step, examines the GIDS-15 in a sample with subclinical depression, a two-armed (intervention group and waiting control group), pragmatic single-blind randomised controlled clinical trial, and a strong external criterion with regard to the sensitivity to change in psychological interventions. We also look at intervention effects between the sexes over time. Since we clearly asked about the biological sex of the participants, we analyse sex differences.

Subclinical depression can be defined as a precursor to major depression ([21]). Since subclinical depression is widespread ([17]), can have far-reaching limitations on life ([45]), and can develop into major depression in the majority of cases ([24]), its investigation is particularly important. Since we are also investigating a screening measure with the GIDS-15, we consider a sample with prodromal symptoms to be sufficient.

This results in the following research questions:(1)What are the psychometric characteristics of the GIDS-15 in this sample?(2)How sensitive to change is the screening version of the GIDS-15 compared to an already established depression measurement instrument?(3)What are the sex differences in the intervention and waiting control groups over time in the total score of the screening version?

## 2. Materials and Methods

### 2.1. Participants and Procedure

The present sample (*N* = 203) was recruited as part of an online training programme for coping with depressive mood ([22]). As part of this prevention study, people with subclinical depression were supported in overcoming their depressed mood themselves by means of an internet-based intervention that promotes their own skills. The six-week training programme with a booster session of four weeks after completion of the last module was designed to enable participants to recognise and cope with stress and crises of everyday life at an early stage before symptoms can develop into clinically relevant depression. In this respect, the programme is classified as indicated prevention. The prevention study was conducted as part of the large-scale EU Innovation Incubator project at Leuphana University Lüneburg in cooperation with the Free University of Amsterdam and Minddistrict Germany. In order to test the effectiveness of the training, a two-armed, pragmatic single-blind randomised controlled clinical trial was implemented with three measurement time points (T0: baseline survey by diagnostic interviews and online questionnaire immediately before randomisation; T1 (post-treatment): seven weeks after baseline by HRSD/QIDS interviews and online questionnaire; T2: three-month follow-up by online questionnaire only). At measurement time T0, a screening was carried out to check whether interested participants met the inclusion criteria of the study. Participants must have had a positive screening for subthreshold depression (CES-D ≥ 16), no major depressive disorder according to DSM IV criteria, an age of 18 years or older, have internet access, not currently receiving psychotherapy or be on a waiting list, not have undergone psychotherapy in the last six months and not be at significant risk of suicide (BDI item 9 > 1). A major depressive episode, a bipolar disorder, a psychotic disorder, and a major depressive episode in the last six months were exclusion criteria. Randomisation was carried out by a researcher not involved in the study using computer-generated numbers and in blocks to ensure equal sample sizes in the groups. This study was conducted in compliance with the Declaration of Helsinki. The data protection regulations were observed. The participants were recruited for the previous study ([9]) of this prevention study via the website of the GesundheitsTraining.Online (GET.ON) project and via the BARMER GEK member magazines. As the desired number of participants had already been reached in the first prevention study ([9]), the project management decided to conduct this follow-up study, as there were still more than 700 applicants on a waiting list to take part in the study. The added value of this study compared to the previous study was the booster session after the last module and the clinical ratings at the post-measurement time. As a result, access for participants in this study was also via the aforementioned website and the member magazines. The study was approved by the Medical Ethics Committee of the University of Lüneburg under the file number Ebert201404_Depr and registered in the German Register for Clinical Studies (No. DRKS00005973). The web-based intervention consisted of six 30 min interactive sessions. These sessions provided psychological interventions based on cognitive–behavioural therapy and problem-solving therapy. An optional refresher session was offered four weeks after completion. The intervention sessions included texts, exercises, personal reports, as well as audio and video clips. During the intervention, participants were supported by an e-coach using adherence-focused guidance. Since the intervention itself is not the main focus of this study, a more detailed description of the training programme, study procedure, and results can be found in the study by [22] ([22]). In the present study, only extracts of the data from the measurement time T0 (here sociodemographic information and data from the anxiety subscale of the Hospital Anxiety and Depression Scale (HADS-A; [61]), Penn State Worry Questionnaire (PSWQ; [7]), Insomnia Severity Index (ISI; [4]), Alcohol Use Disorders Identification Test (AUDIT; [50]), Quick Inventory of Depressive Symptomatology-Clinican Rating (QIDS-CR 16; [47]) and Hamilton Rating Scale for Depression (HRSD-24; [47]) were used. In addition, we used the available data from the Gender-specific binary depression screening version (GIDS-15; [39]) and the Center for Epidemiological Studies Depression Scale (CES-D; [40]) from all measurement times, T0, T1, and T2.

The sample characteristics of this study can be found in Table 1. In descriptive terms, the intervention and waiting control groups do not differ in terms of the characteristics listed. However, among both groups, only one-fifth of participants are men. Overall, the people in both groups are on average in their mid-40s, the majority are married or in a partnership, highly educated, and in employment.

### 2.2. Assessments

#### 2.2.1. Gender-Specific Binary Depression Screening Version (GIDS-15; [39])

The Gender-specific binary depression screening version (GIDS-15) was developed from the original Gender Inclusive Depression Scale (GIDS) by [31] ([31]). The GIDS was translated and analysed in two large German-speaking mixed-sex samples in terms of factor analysis, psychometric parameters, and sex and age effects ([39]). Item reduction resulted in the GIDS-15, which contains a total of 15 items that, based on the original version, are question-dependent on a two-stage (‘true’, ‘false’), a three-stage (‘yes’, ‘no’, ‘I don’t know’), a four-stage (‘often’, ‘sometimes’, ‘rarely’, ‘never’) and a five-stage (‘all the time’, ‘most of the time’, ‘some of the time’, ‘a little’, ‘not even once’) response format. The majority of the questions are to be assessed using the three-stage response format. The total value of the GIDS-15 is calculated by adding up the point values in the factors. However, each person only receives a maximum of one point per factor, regardless of whether they agree with more than one item of the factor. In the first validation study ([39]), five factors were extracted by factor analysis: conventional depressive symptoms, stress perception, anxiousness, aggressiveness, substance use. The internal consistencies in an online sample were 0.85 (Cronbach’s alpha) for the overall scale and 0.87 (Cronbach’s alpha) for the factor conventional depressive symptoms, 0.72 (Spearman–Brown coefficient) for the factor anxiety, 0.70 (Spearman–Brown coefficient) for the factor stress perception, 0.53 (Spearman–Brown coefficient) for the factor aggressiveness and 0.51 (Spearman–Brown coefficient) for the factor substance use. As expected, the construct validity was confirmed by means of correlation calculations with other methods. Further results and explanations of the sum score formation can be found in the first validation study ([39]).

#### 2.2.2. Additional Assessments

##### Center for Epidemiological Studies for Depression Scale (CES-D; [40])

The Center for Epidemiological Studies Depression Scale (CES-D) is a short self-report scale consisting of 20 items that was originally developed to measure depressive symptoms in large-scale epidemiological studies in the general population. Typical affective, cognitive, somatic, and social symptoms of depression during the past week are assessed. A high internal consistency was found in the general population (Cronbach’s alpha 0.85; [40]). For the current overall sample, an acceptable internal consistency (Cronbach’s alpha) of 0.78 was found at measurement time 0, a high internal consistency (Cronbach’s alpha) of 0.87 at measurement time 1, and a high internal consistency (Cronbach’s alpha) of 0.91 at measurement time 2.

##### Anxiety Subscale of the Hospital Anxiety and Depression Scale (HADS-A; [61])

With 14 items, the Hospital Anxiety and Depression Scale (HADS) by [61] ([61]) is a brief screen for depression (seven items) and anxiety (seven items). The items do not refer to severe psychopathological symptoms, so the scale can also be used for milder forms of disorder or for use in the general population. In this study, the anxiety subscale was used. An acceptable internal consistency (Cronbach’s alpha) of 0.72 was reported for this subscale in a standardisation study of the HADS on a sample that can be considered representative of the general population in Germany ([27]). For the current total sample, we calculated a critical internal consistency (Cronbach’s alpha) of 0.67 at measurement time 0.

##### Penn State Worry Questionnaire (PSWQ; [7])

The Penn State Worry Questionnaire (PSWQ) is designed to measure pathological worry. The ultra-brief version with three items was used here. The validation study on clients with panic disorder with/without agoraphobia, social anxiety disorder, or obsessive–compulsive disorder showed an internal consistency (Cronbach’s alpha) of 0.85 ([7]). In a sample of Dutch psychology students, a Cronbach’s alpha of 0.91 was found ([54]). For the current total sample, we calculated a high internal consistency (Cronbach’s alpha) of 0.84 at measurement time 0.

##### Insomnia Severity Index (ISI; [4])

The Insomnia Severity Index (ISI) is a short self-report instrument with seven items to measure the person’s perception of insomnia. In a German-speaking random sample recruited with the offer to participate in a sleep training group or an online self-help programme, Cronbach’s alpha was a good 0.83 ([19]). For the current total sample, we calculated a high internal consistency (Cronbach’s alpha) of 0.86 at measurement time 0.

##### Alcohol Use Disorders Identification Test (AUDIT; [50])

The Alcohol Use Disorders Identification Test (AUDIT) is a self-report-based screening procedure that can be used to identify people with high alcohol consumption or hazardous drinking habits. In the general population, a Cronbach’s alpha of 0.75 ([46]) shows a moderate internal consistency. For the current overall sample, an acceptable internal consistency (Cronbach’s alpha) of 0.79 was found at measurement time 0.

##### Quick Inventory of Depressive Symptomatology-Clinican Rating (QIDS-CR 16; [47])

The Quick Inventory of Depressive Symptomatology-Clinican Rating (QIDS-CR 16) uses 16 items to assess the nine depressive symptom areas according to DSM-IV during the last seven days. To the best of our knowledge, no study has investigated the internal consistency of the QIDS-CR 16 in an online sample with subclinical depression. In a sample of individuals with major depression, the QIDS-CR 16 showed a high internal consistency of 0.85 ([55]). The interrater reliability in our study was ensured by an independent, experienced diagnostician and was 0.97 based on data from 10% of the participants ([22]).

##### Hamilton Rating Scale for Depression (HRSD-24; [47])

The Hamilton Rating Scale for Depression (HRSD-24) is a widely used 24-item scale for measuring depression that is rated by clinicians. In a study of German-speaking individuals with a diagnosis of depression or dysthymia who were recruited for a randomised trial of an online prevention programme, the internal consistency of the HRSD-24 was a good 0.76 Cronbach’s alpha ([44]). In a sample of people with chronic major depression, the Cronbach’s alpha was a very good 0.88 ([47]). In our study, the interrater reliability was 0.94 ([22]).

### 2.3. Statistical Analyses

A general data screening was performed on the raw data of the sample. One person who stated their sex as diverse was excluded (*n* = 1), because our study considered only binary sex. Due to drop-outs, because the intervention was not completed, the number of participants dropped from 203 at measurement time T0 to 178 at measurement time T1 and 162 at measurement time T2. Systematic analyses of the reasons for drop-out were not carried out because the participants could not be reached. We determined the reliability using Cronbach’s alpha for the total score of the GIDS-15, separated into the intervention and the waiting control group, for the different measurement time points. Multitrait-multimethod analyses (MTMM analyses) were conducted to test whether the screening version of the GIDS-15 is valid ([11]). The GIDS-15 was compared with the CES-D and the two external ratings QIDS-CR-16 and HRSD-24, as well as with the HADS-A, PSWQ, ISI, and AUDIT. To demonstrate construct validity, the correlations in the correlation matrix are assessed by pairwise comparisons to determine whether the criteria of convergent and discriminant validity are met. If not all criteria are completely fulfilled, this does not necessarily argue against the construct validity. It is expected that the GIDS-15, the CES-D, and the two clinician ratings QIDS-CR-16 and HRSD-24 (heteromethod), which measure the same construct depression (monotrait), will show significant positive correlations. This would be evidence of convergent validity. In contrast, low correlations are expected between measurement methods that measure different constructs (multitrait) (e.g., depression and alcohol consumption), both within (monomethod) and between the methods (multimethod). This would be evidence for the discriminant validity. In summary, the following criteria should therefore be met to demonstrate the construct validity of the GIDS-15: (a) To demonstrate convergent validity, the correlations between monotrait and multimethod should be significantly different from zero and as high as possible. (b) To prove discriminant validity, the correlations between heterotrait and monotrait should be smaller than the correlations between monotrait and multimethod, and (c) the correlations between heterotrait and multimethod should be smaller than the correlations between monotrait and multimethod. To test the sensitivity of change in the GIDS-15, we also conducted effect size comparisons using Cohen’s d between the GIDS-15 and the CES-D separately for the intervention and wait-list control groups, between T0 and T1, and between T0 and T2. We used a T-test for paired samples. The evaluation of the characteristic values is based on [15]’s ([15]) rule of thumb, according to which effect sizes with a value of 0.20 are described as small, 0.50 as medium, and 0.80 as large. To test for differences in the change in the GIDS-15 and the CES-D over time, we conducted a z-test for paired samples. The development of the GIDS-15 over time was analysed with a linear mixed model. We included the intervention group, sex, time, and all possible interaction terms, including three-way interactions, into the model as fixed effects and participants as a random effect. The significance of interaction effects was evaluated by comparison with the significance level of 0.05. We report regression coefficients and the marginal means with the corresponding 95% confidence interval. All statistical analyses were carried out using the SPSS version 27 programme (Statistical Package for Social Sciences) for Windows, and Stata version 18 ([53]).

## 3. Results

### 3.1. Research Question 1

In terms of reliability, Cronbach’s alpha generally showed few differences between the intervention and the waiting control group. However, there were differences between the individual measurement times. While the internal consistencies (Cronbach’s alpha) were >0.60 at T0, they were >0.70 at T1 and T2 (Table 2). Due to the heterogeneity of the GIDS-15, we consider the calculated parameters to be sufficient at T0 and acceptable at T1 and T2.

Table 3 shows the Pearson correlations for the GIDS-15 with the various other methods for the total sample. There were significant positive correlations between the GIDS-15, the CED-S, the QIDS-CR 16, and the HRSD-24 (monotrait-heteromethod correlations). All validity coefficients with values between 0.34 and 0.81 are significantly different from zero and relevant in size, so that convergent validity can be considered proven. The convergent validity between the two external assessment procedures is particularly high (r = 0.81). In contrast, the convergent validities of the GIDS-15 with the other procedures are lower (r ≥ 0.34). The correlations between different traits measured by the same method (heterotrait monomethod coefficients) should be lower than the corresponding convergent validity coefficients of these traits. For example, the correlation between the GIDS-15 and the HADS-A is lower (r = 0.20) than the correlations just reported. While the correlations with the other survey instruments (heterotrait) for the GIDS-15 are all lower (r = 0.10 to 0.29) than the previous correlations (monotrait), three out of 14 correlations in the overall matrix do not fulfil this requirement (r = −0.14 to 0.54). This means that the first criterion of discriminant validity is not completely fulfilled. In addition, the coefficients of the heterotrait-heteromethod correlations should be lower than the heterotrait-monomethod correlations. This requirement applies to almost none of the correlation coefficients mentioned. The correlations are between r = 0.00 and 0.40, meaning that the coefficients do not fulfil the second criterion of discriminant validity. Overall, the analysis of the MTMM matrix according to the Campbell–Fiske criteria provides clear indications of the convergent validity of the GIDS-15, while the discriminant validity criteria can only be partially fulfilled.

### 3.2. Research Question 2

We conducted effect size comparisons using Cohen’s d between the GIDS-15 and the CES-D separately for the intervention and wait-list control groups, between T0 and T1, and between T0 and T2. We used a T-test for paired samples. The results are shown in Table 4. In the intervention group, large effects according to [15] ([15]) were shown between the measurement times T0 and T1 as well as between T0 and T2 in both procedures. The Cohen’s d values in the CES-D were higher than those of the GIDS-15. In the waiting control group, both comparisons of the measurement times and both procedures showed almost medium effects according to [15] ([15]). The Cohen’s d values did not differ between the two measurements.

### 3.3. Research Question 3

In the linear mixed model, we found a three-way interaction between intervention group, sex, and time (*p* = 0.017). This means that the intervention effect over time differs between men and women. While the intervention effect for women is relatively constant over time, for men, an increase in the intervention effect was observed (Figure 1). The estimated average GIDS-15 score for men at the last time point differed substantially between the intervention group (marginal mean = 1.31, 95% CI: 0.62–2.00) and the control group (marginal mean = 3.18, 95% CI: 2.55–3.81). For women, the intervention effect was less pronounced at the last time point (Intervention group: marginal mean = 2.08, 95% CI: 1.74–2.42; Control group: marginal mean = 2.76, 95% CI: 2.45–3.06). The individual coefficients of the linear mixed model can be found in Appendix A.

## 4. Discussion

The aim of the present study was to further validate a screening version for gender-specific depression diagnosis in Germany using the GIDS-15 in order to confirm findings from the preliminary validation ([39]) and, for example, to extend them to include the aspect of change sensitivity. The first validation study included people with problematic alcohol consumption ([39]). Therefore, the screening version was now used in a sample with subclinical depression, which was considered relevant for the scope of the measurement instrument, as part of a two-armed, pragmatic, single-blind, randomised, controlled clinical trial. In this study, clinical ratings were available and thus stronger, valid external criteria than only self-evaluation procedures as in the first validation step ([39]). We also investigated the intervention effects on women and men in order to derive possibly more adequate treatment options.

### 4.1. Research Question 1

With regard to the measured psychometric parameters, the internal consistency, measured with Cronbach’s alpha, achieved in this sample at best sufficient values for the GIDS-15 at measurement time T0. Acceptable values for Cronbach’s alpha could be calculated at measurement times T1 and T2. It is possible that at T0, the measurement time point before randomisation, the participants’ awareness of their subsequent assignment to the intervention or wait-list control group influenced their response behaviour, which in turn would have an impact on reliability (see Limitations). In addition, the lower internal consistency at this measurement time point in this sample compared to a general population sample could be due to the heterogeneity of subclinical depression, which leads to greater response variability. Overall, the internal consistencies of the GIDS-15 were below those of the first validation study (there in sample 1: Cronbach’s alpha: 0.85; in sample 2: Cronbach’s alpha: 0.81; [39]) for the possible reasons mentioned, but at a comparable level at measurement times T1 and T2.

In our study, we found clear evidence for the convergent validity of the GIDS-15 with other established measurement instruments, while the discriminant validity was only partially fulfilled. The comparable level of the heterotrait monomethod coefficients and the heterotrait heteromethod coefficients in MTMM analyses indicates that there could be method effects that could lead to a falsification of the results. The limitations in construct validity could be due to the sufficient reliability at measurement time T0 only because the correlations with other measurement instruments are lower. At the same time, the construct of depression is measured imprecisely as a result. However, correlations with other methods designed to measure depression are solid, which indicates a convergent construct validity of the GIDS-15. To the best of our knowledge, validation studies of previous gender-specific depression instruments in Germany have only used self-assessment instruments so far ([33]; [34]). It is therefore a significant strength of this study that the GIDS-15 was validated with a strong external criterion based on clinical ratings ([39]). The fact that the correlation levels are only moderate could be due to the heterogeneous structure of the GIDS-15. In addition to the classic symptoms of depression, it also measures other constructs (such as stress perception) that are not included in the measurement instruments used in this study. The special characteristics of the sample could also have an influence on the moderate correlation levels. The sample is one with subclinical depression, meaning that the range of possible symptoms is greater than in people with manifest depression. The criteria for discriminant validity are only partially fulfilled. The questionnaires used in this study and thus the constructs surveyed correlate positively and substantially with each other. The vast majority of the questionnaires used here measure constructs that are associated with depressive symptoms. For example, worry and sleep disorders can be symptoms of depression. According to the current classification systems, ICD and DSM, anxiety is not directly a symptom of depression, but there is a high comorbidity of up to 60% ([29]). In this respect, it can be assumed that the questionnaires used here, even though they were designed for different symptom areas, capture something in common. Therefore, the detection of discriminant validity could be more difficult with these questionnaires. Only the correlations with the AUDIT, which measures alcohol-related complaints that are not related to depression symptoms, indicate discriminant validity for the GIDS-15.

### 4.2. Research Question 2

When investigating the sensitivity to change in questionnaires, one of the fundamental questions is whether the results found are due to the sensitivity to change in the measurement instrument used or to the effectiveness of the intervention ([28]). Based on several previous findings on the interventions used in this study, which have shown effectiveness in various samples ([22]; [8], [9]), we strongly assume that the interventions are generally effective and that we can therefore assess the change sensitivity of the GIDS-15. In terms of examining the change sensitivity of the GIDS-15 across measurement time points, the GIDS-15 was found to be sensitive to change for the applied interventions. The sensitivity to change was observed in two comparisons of measurement time points: firstly, in the comparison before randomisation and after the intervention, and secondly, before randomisation and during the follow-up period. The extent of changes in the intervention group shown in the GIDS-15 was roughly comparable to the results found in other studies on the effectiveness of online interventions for subclinical depression. In an online prevention study by [8] ([8]), in which people with subclinical depression also took part, a large effect size was found within the group according to Cohen’s d. In another study with people aged 50 years and older with subclinical depression and an online intervention programme without professional support, a Cohen’s d of 1.00 was calculated ([51]). At the same time, the changes in the GIDS-15 are lower in the waiting control group than in the intervention group. However, the results can only be categorised in comparison with similar measurement instruments ([28]). The effect sizes found are comparable with the already established measurement instrument (CES-D), which was used at the same time. However, the CES-D is more sensitive to changes compared to the intervention group. It recorded the changes caused by the psychological interventions in the study more strongly. One reason for the comparatively lower sensitivity to change could again be the design of the GIDS-15, which records externalising behaviours that were not conceptually part of the interventions. However, it is also possible that the sample characteristics were unfavourable for the study conducted (see limitations). Due to the sex distribution (significantly fewer men in both the intervention and the waiting control group) and the necessary condition of subclinical depression, the effects of sensitivity to change could also be capped. In connection with the answer to research question 1, our findings therefore provide further and extended validation evidence in a sample population relevant to the area of application, which is required (e.g., that the measurement instrument is also valid for change in the application of psychological interventions and is validated with a strong external criterion; [39]).

### 4.3. Research Question 3

In our study, we find a three-way interaction between the intervention group, sex, and time. Accordingly, we observe an increase in the intervention effect over time for men, which is not the case for women. This result can be discussed in the context of treatment outcomes that are influenced by the sex of clients. The DSM-5-TR ([3]) assumes, based on research findings, that women and men differ in their depressive symptom patterns. For example, it states that men with depression are more likely to use maladaptive problem-solving strategies. As the psychological interventions in this study also include elements from problem-solving therapy, the interaction effect found in men could therefore be evidence that an adequate aspect could be addressed by the interventions and also stabilised and increased over time, as would be expected with a successful expansion of problem-solving skills ([20]). At the same time, the interventions may not have been as helpful for the women as they were for the men. If this explanation is followed, future studies on web-based interventions should therefore take into account previous analyses on gender-specific determinants and patterns of online health information searches ([5]) and the consideration of gender stereotypes in the design and perception of digital products ([6]) in order to address sex- and gender-specific aspects in the use and effectiveness of such offers and to optimise their effectiveness for different user groups. And this is simply because we generally see that men are significantly less likely to take advantage of prevention services in the area of mental health ([59]; [10])—as is also the case in this online prevention study. However, our results are in contrast to findings according to which no differences in effectiveness between men and women are found in psychological interventions for the prevention of depression ([26]). And an individual participant data meta-analysis of randomised controlled trials of internet-based interventions for the prevention of depression also showed that sex was not a moderator of the effects found ([41]). In this respect, the effects observed in our study could also be due to the construction and interpretation of the GIDS-15 in conjunction with the small number of men in the study. The total score results from the sum of the individual subscales, whereby a maximum of one point is awarded for each subscale, regardless of how many items are confirmed ([39]). Due to the significantly lower number of men in our study, even small changes in their responses could contribute to a larger variance and thus increase the probability of detecting an intervention effect. An indicator for our considerations is the confidence intervals in the different groups and at the different measurement times, which are significantly larger for men than for women. The standard error was wider in the male group because it contained fewer individuals. In this respect, it might be worth conducting our study on a larger sample with a comparable sex ratio to counter selection bias (see limitations).

### 4.4. Limitations

The study has some limitations. Some of the following limitations could be addressed by adjusting the changes in the GIDS-15 and using more suitable samples. The data collection for the study took place in 2014 and 2015. Due to the existing research gap in well-validated gender-sensitive depression questionnaires in Germany ([62]; [33]), we still consider the publication of the data to be meaningful. Nevertheless, it would be interesting to examine the GIDS-15 on a current sample, as disease perception and impact may have changed due to the COVID-19 pandemic. This could be an important future research approach. Due to the study design, possible selection and recruitment effects must be considered. In particular, the data quality before randomization could have been affected by participants’ expectations that they could influence their allocation to the intervention or waiting control group. This could explain the only adequate reliability of the GIDS-15 and other measurements at this measurement point. The CES-D is not an optimal measuring instrument, as it is mixed with anxiety symptoms. A comparison of the GIDS-15 with the PHQ-9 ([52]) would be desirable. At the same time, however, the GIDS-15 also includes a factor that measures anxiety. The unequal sex distribution and the necessary requirement of having only a subclinically expressed depression could have influenced the variability and thus the variance. This has implications for the (non-)discovery of statistical effects. Therefore, the studies should be conducted again with larger sample sizes and comparable sex ratios. In the study, we clearly and exclusively asked about the biological sex of the participants. Future research in this area should take a more differentiated approach to gender, including, for example, social gender ([18]), as it is assumed that the expression of symptoms is linked to social gender ([16]). Further gender diversity should also be taken into account in the future, as purely binary research appears outdated ([25]).

## 5. Conclusions

In order to adequately take the gender-dependent manifestations of depression in clinical practice into account, suitable measurement instruments are essential. In the present study, the Gender-specific binary depression screening version (GIDS-15) proved to be sufficiently reliable and construct-valid in a second validation step on a relevant sample with a strong external criterion. As it has a comparable sensitivity to change as an already established measurement instrument, the GIDS-15 appears to be a sensible and economical option for use in the follow-up and longitudinal diagnosis of depressive symptoms and in primary care. However, it refers to the gender binary. Since intervention effects are questionable, this instrument would have to be validated in other samples and possibly adapted slightly.

## Figures and Tables

**Figure 1 behavsci-15-01253-f001:**
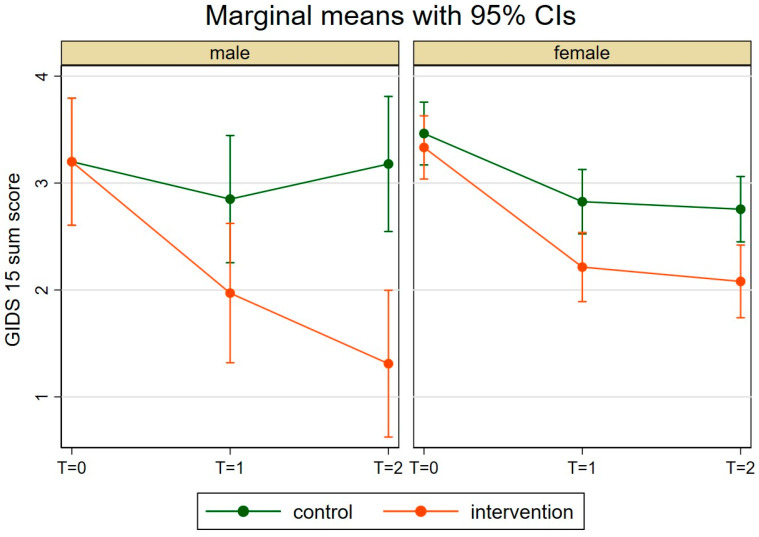
Linear mixed model investigating the development of the GIDS-15 over time. Note. CI = Confidence interval, T = measurement time.

**Table 1 behavsci-15-01253-t001:** Sample characteristics of intervention and control group in terms of sex, age, level of education, relationship status, and employment status at baseline. Continuous variables are presented as mean (standard deviation), while categorical variables are presented as absolute and (relative) frequencies (modified from publication from [22]).

Characteristic	Intervention Group (N = 101)	Control Group (N = 102)	Total Sample (N = 203)
Sex			
Men	20 (19.8%)	20 (19.6%)	40 (19.7%)
Women	81 (80.2%)	82 (80.4%)	163 (80,3%)
Age	44.65 (11.71)	43.75 (11.84)	44.20 (11.75)
Relationship			
Single	26 (25.7%)	28 (27.5%)	54 (26.6%)
Married or cohabiting	65 (64.4%)	53 (52.0%)	118 (58.1%)
Divorced or separated	9 (8.9%)	20 (19.6%)	29 (14.3%)
widowed	1 (1.0%)	1 (1.0%)	2 (1.0%)
Level of education			
Low (primary)	1 (1.0%)	3 (2.9%)	4 (2.0%)
Middle (secondary)	16 (15.8%)	16 (15.7%)	32 (15.8%)
High (A-level or higher)	84 (83.2%)	83 (81.4%)	167 (82.3%)
Employment status			
Employed	89 (88.1%)	87 (85.3%)	176 (86.7%)
Unemployed or seeking work	2 (2.0%)	4 (3.9%)	6 (3.0%)
On sick leave	0 (0%)	2 (2.0%)	2 (1.0%)
Non-working	10 (9.9%)	9 (8.8%)	19 (9.4%)

**Table 2 behavsci-15-01253-t002:** Reliabilities (Cronbach’s alpha) of the GIDS-15 for the different measurement times and the respective groups.

Measurement Time	Group	Cronbach’s Alpha
T0	Complete sample	0.61
	IG	0.60
	WG	0.63
T1	Complete sample	0.74
	IG	0.75
	WG	0.71
T2	Complete sample	0.74
	IG	0.75
	WG	0.71

Note. Measurement time: T0 = baseline before randomisation, T1 = seven weeks after baseline, T2 = three-month follow-up, IG = intervention group, WG = waiting control group, GIDS-15 = Gender-specific binary depression screening version.

**Table 3 behavsci-15-01253-t003:** MTMM matrix at measurement time 0 for the entire group (n = 203).

	Self-Assessment						Clinician Ratings	
Self-Assessment	GIDS-15	CES-D	HADS-A	PSWQ	ISI	AUDIT	QIDS-CR 16	HRSD-24
GIDS-15	1							
CES-D	0.47 **	1						
HADS-A	0.20 **	0.37 **	1					
PSWQ	0.29 **	0.51 **	0.54 **	1				
ISI	0.15 *	0.28 **	0.13	0.20 **	1			
AUDIT	0.10	−0.06	−0.06	−0.14 *	−0.00	1		
Clinician ratings								
QIDS-CR 16	0.41 **	0.46 **	0.34 **	0.33 **	0.39 **	0.06	1	
HRSD-24	0.34 **	0.48 **	0.39 **	0.40 **	0.32 **	0.00	0.81 **	1

Note. GIDS-15 = Gender-specific binary depression screening version, CES-D = Center for Epidemiological Studies for Depression Scale, HADS-A = Anxiety subscale of the Hospital Anxiety and Depression Scale, PSWQ = Penn State Worry Questionnaire, ISI = Insomnia Severity Index, AUDIT = Alcohol Use Disorders Identification Test, QIDS-CR 16 = Quick Inventory of Depressive Symptomatology-Clinican Rating, HRSD-24 = Hamilton Rating Scale for Depression, **: Correlation is significant at the 0.01 level (2-sided); *: Correlation is significant at the 0.05 level (2-sided).

**Table 4 behavsci-15-01253-t004:** Effect size comparisons (Cohen’s d).

Group	Comparison of the Points in Time	GIDS-15 [95% CI]	CES-D [95% CI]	*p*-Value
IG	T0-T1	0.80 [0.55–1.05]	1.19 [0.91–1.48]	0.007
IG	T0-T2	0.87 [0.59–1.14]	1.12 [0.82–1.411]	0.091
WG	T0-T1	0.40 [0.19–0.61]	0.54 [0.33–0.75]	0.197
WG	T0-T2	0.40 [0.18–0.61]	0.39 [0.18–0.61]	0.955

Note. GIDS-15 = Gender-specific binary depression screening version, CES-D = Center for Epidemiological Studies for Depression Scale, CI = Confidence interval, IG = intervention group, WG = waiting control group, T0 = measurement time 0, T1 = measurement time 1, T2 = measurement time 2.

## Data Availability

The datasets presented in this article are not readily available. Requests to access anonymized datasets should be directed to the corresponding author.

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
