# Peer review of "Further Validation Study of the Gender-Specific Binary Depression Screening Version (GIDS-15) and Investigation of Intervention Effects"

_behavsci, 2025, doi:10.3390/bs15091253_

Round 1
Reviewer 1 Report
Comments and Suggestions for Authors
The male sample is much smaller than the female sample, so comparisons are impaired to support the conclusions that are shown and argued as the core of the value of the article (Abstract – Conclusions)
It can be found (2.1 Participants and procedures; 4 Discussion - limitations) that the sample comes from far in the past (2014-2015), so the advent of the COVID-19 pandemic may have changed much of the picture.
Author Response
Thank you very much for your supportive comments on our study. We have carefully considered both of your suggestions for improvement and would like to respond briefly.
- The male sample is much smaller than the female sample, so comparisons are impaired to support the conclusions that are shown and argued as the core of the value of the article (Abstract – Conclusions)
Response: We have revised the abstract to better classify our results and to mention limitations at this point: “It can be used for progression and intervention diagnostics, although the intervention effect that was found can only be interpreted to a limited extent due to significant sample size differences between men and women.“ (page 1, Abstract – Conclusions)
- It can be found (2.1 Participants and procedures; 4 Discussion - limitations) that the sample comes from far in the past (2014-2015), so the advent of the COVID-19 pandemic may have changed much of the picture.
Response: We also consider the second point to be extremely important and thank you for mentioning it. It greatly enriches our article. As we have not yet found any specific studies in our literature search, we have decided to add a generic supplement to the limitations: “The data collection for the study took place in 2014 and 2015. Due to the existing research gap in well-validated gender-sensitive depression questionnaires in Germany (Zülke et al., 2018; Möller-Leimkühler et al., 2022), we still consider the publication of the data to be meaningful. Nevertheless, it would be interesting to examine the GIDS-15 on a current sample, as disease perception and impact may have changed due to the COVID-19 pandemic. This could be an important future research approach.“ (page 14, subheading Limitations, third sentence)
We hope that the changes adequately reflect your points.
Thank you again and best regards,
The authors
Reviewer 2 Report
Comments and Suggestions for Authors
Thanks to authors, very good article.
Some small remarks:
- Will be nice to have little bit more information about excluded patients form the survey (162 from 203 in the visit T0)
- Little bit more about recruited patients and online training programme - how the patients apply to this group, are there very severe depressive patients or only subclinical, mild depressive patients? Does severity of patients impact results of your research?
- Page 12, 5th line - the authors speak about "depressive symptoms" and mentioned "anxiety, worry". Anxiety disorder seams to be separate a disorder and anxiety is not a symptom of depression (ICD10 and DSM 5). Depression and anxiety comorbidity is high - about 60%. Please clarify.
Author Response
We would like to thank you very much for your very helpful comments on our study. We have looked closely at the suggestions for improvement and have implemented all of them in a revised version of our publication.
Below you will find a detailed point-by-point response to all your comments. In addition, we have submitted the revised manuscript.
- Will be nice to have little bit more information about excluded patients form the survey (162 from 203 in the visit T0)
Response: We have added a sentence on the reasons for drop-outs. The revised manuscript states: “Due to drop-outs, because the intervention was not completed, the number of participants dropped from 203 at measurement time T0 to 178 at measurement time T1 and 162 at measurement time T2. Systematic analyses of the reasons for drop-out were not carried out because the participants could not be reached.“ (page 7, subheading Statistical Analyses, third sentence)
2. Little bit more about recruited patients and online training programme - how the patients apply to this group, are there very severe depressive patients or only subclinical, mild depressive patients? Does severity of patients impact results of your research?
Response: We have added a paragraph defining subclinical depression, which we hope incorporates your comments. In addition, we believe that the methods section (page 3 and ongoing) contains a number of answers to your questions, e.g., regarding recruitment or the distinction between subclinical and clinical depression in the study's inclusion and exclusion criteria which hopefully answers the above questions. The paragraph defining subclinical depression reads: “Subclinical depression can be defined as a precursor to major depression (Eaton et al., 1995). Since subclinical depression is widespread (Cuijpers et al., 2004), can have far-reaching limitations on life (Rucci et al., 2003), and can develop into major depression in the majority of cases (Frank et al., 1991), its investigation is particularly important. Since we are also investigating a screening measure with the GIDS-15, we consider a sample with prodromal symptoms to be sufficient.“ (page 3, subheading Aims of the current study, second paragraph)
3. Page 12, 5th line - the authors speak about "depressive symptoms" and mentioned "anxiety, worry". Anxiety disorder seams to be separate a disorder and anxiety is not a symptom of depression (ICD10 and DSM 5). Depression and anxiety comorbidity is high - about 60%. Please clarify.
Response: We took the comment as an opportunity to present our findings in a more differentiated manner. You will find the comments in the appropriate place: “The vast majority of the questionnaires used here measure constructs that are associated with depressive symptoms. For example, worry and sleep disorders can be symptoms of depression. According to the current classification systems ICD and DSM, anxiety is not directly a symptom of depression, but there is a high comorbidity of up to 60% (Kaufman & Charney, 2000). In this respect, it can be assumed that the questionnaires used here, even though they were designed for different symptom areas, capture something in common.“ (page 12, subheading Discussion – Research question 1), last paragraph)
We hope that the changes adequately reflect your points.
Thank you again and best regards,
The authors
Reviewer 3 Report
Comments and Suggestions for Authors
First, I commend you for the effort in facing a complicated task. It’s very valuable in the clinical field and has led to a lot of reflection and learning. So, aside from this research, I’m looking forward to you continuing to work on this subject. I have made a few observations to the paper, hoping they will be of use to you. Because the copy of the manuscript available did not have line numbering, my comments refer to sections.
Introduction:
Page 2, second paragraph: authors mention the inclusión of irritability as a criterion included in CIE 11 for depression. Irritability has been included in the main description of depression in the DSM system a long time ago, and as a symptom that can replace the depressive mood in children and adolescents. Although it has not been included as a main criterion for adults, it has been mentioned in the description since the DSM-IV was published. For more clarity, authors should mention the difference: it was a main criterion only for child and adolescent depression in DSM IV system, while it wasn’t for adults, being mentioned only as a complementary description. And more recently, CIE 11 included it as a main criterion for adult depression.
Page 3 Aims of the current study
The authors said that the sample included people with subclinical depression. I think you should bring your definition of subclinical depression for the readers' understanding here, and how the subclinical depression was determined, as well as the reasons you decided to test the scale in subclinical depression, instead of clinical depression.
Discussion
About sex and or gender: dear authors, it’s always a bit dilemmatic to choose what you will look for, if sex or gender, or both. However, I believe it's essential to include gender on the screen, given the sample composition (only one-fifth were males), as gender significantly influences many of the differences you are looking for. It’s widely known that men attend medical consultations less frequently than women, especially in the realm of psychological symptoms. So, even when I understand that you asked specifically for biological sex, given the fact that the scale refers to gender, and gender is always a socially constructed part of identity, I suggest you use the word gender instead of sex in the discussion.
Author Response
Thank you very much for your supportive comments on our study. We have carefully considered your suggestions for improvement and would like to respond briefly.
- Introduction:
Page 2, second paragraph: authors mention the inclusión of irritability as a criterion included in CIE 11 for depression. Irritability has been included in the main description of depression in the DSM system a long time ago, and as a symptom that can replace the depressive mood in children and adolescents. Although it has not been included as a main criterion for adults, it has been mentioned in the description since the DSM-IV was published. For more clarity, authors should mention the difference: it was a main criterion only for child and adolescent depression in DSM IV system, while it wasn’t for adults, being mentioned only as a complementary description. And more recently, CIE 11 included it as a main criterion for adult depression.
Response: Thank you for your comment. We have gratefully included this aspect in the relevant section of our study: “In the recently introduced eleventh version of the International Statistical Classification of Diseases and Related Health Problems (ICD-11; WHO), irritability is also included as a possible affective component of depression in adults in the disorder description, which could facilitate the identification of depressive disorders in men by clinicians or in primary care (Østergaard et al., 2023). It should be noted that irritability was already listed in the DSM-IV as a major criterion for depression in children and adolescents, while it was not considered as such in adults (APA, 2000).“ (page 2, subheading Introduction, second parapgraph)
2. Aims of the current study:
The authors said that the sample included people with subclinical depression. I think you should bring your definition of subclinical depression for the readers' understanding here, and how the subclinical depression was determined, as well as the reasons you decided to test the scale in subclinical depression, instead of clinical depression.
Response: We have taken your comments on subclinical depression as an opportunity to add a paragraph on the definition, effects, and selection for our study to the objectives of our investigation. Please find our comments in the relevant section: “Subclinical depression can be defined as a precursor to major depression (Eaton et al., 1995). Since subclinical depression is widespread (Cuijpers et al., 2004), can have far-reaching limitations on life (Rucci et al., 2003), and can develop into major depression in the majority of cases (Frank et al., 1991), its investigation is particularly important. Since we are also investigating a screening measure with the GIDS-15, we consider a sample with prodromal symptoms to be sufficient.“ (page 3, subheading Aims of the current study, second paragraph)
3. Discussion:
About sex and or gender: dear authors, it’s always a bit dilemmatic to choose what you will look for, if sex or gender, or both. However, I believe it's essential to include gender on the screen, given the sample composition (only one-fifth were males), as gender significantly influences many of the differences you are looking for. It’s widely known that men attend medical consultations less frequently than women, especially in the realm of psychological symptoms. So, even when I understand that you asked specifically for biological sex, given the fact that the scale refers to gender, and gender is always a socially constructed part of identity, I suggest you use the word gender instead of sex in the discussion.
Response: Even among the authors, we had many discussions about whether we should refer to sex or gender in our work. Although we fully agree with your arguments, we would like to retain our previous wording. Yes, it is true that the GIDS-15 is based on social gender roles and corresponding role expectations and behaviors. However, since we only asked about the biological sex of the participants in the population studied, it seems only logical to us to also refer to sex, even though the GIDS-15 does, of course, cover aspects of gender.
We hope that the changes adequately reflect your points.
Thank you again and best regards,
The authors
Round 2
Reviewer 1 Report
Comments and Suggestions for Authors
On my opinion, the present version has been
sufficiently improved to warrant publication.